# The Phase Diagram and Exotic Magnetostrictive Behaviors in Spinel Oxide Co(Fe_1−*x*_Al_*x*_)_2_O_4_ System

**DOI:** 10.3390/ma12101685

**Published:** 2019-05-23

**Authors:** Chao Zhou, Azhen Zhang, Tieyan Chang, Yusheng Chen, Yin Zhang, Fanghua Tian, Wenliang Zuo, Yang Ren, Xiaoping Song, Sen Yang

**Affiliations:** 1School of Science, MOE Key Laboratory for Nonequilibrium Synthesis and Modulation of Condensed Matter, State Key Laboratory for Mechanical Behavior of Materials, Xi’an Jiaotong University, Xi’an 710049, China; chao.zhou@xjtu.edu.cn (C.Z.); zhangazhen@stu.xjtu.edu.cn (A.Z.); yzhang18@xjtu.edu.cn (Y.Z.); zuowenliang@xjtu.edu.cn (W.Z.); tfh2017@xjtu.edu.cn (F.T.); songxp@mail.xjtu.edu.cn (X.S.); 2ChemMatCARS, the University of Chicago, Argonne, IL 60439, USA; changty.6@stu.xjtu.edu.cn (T.C.); yschen@cars.uchicago.edu (Y.C.); 3X-Ray Science Division, Advanced Photon Source, Argonne National Laboratory, Argonne, IL 60439, USA; ren@aps.anl.gov

**Keywords:** magnetostriction, spinel oxide, phase boundary, phase diagram

## Abstract

We report the magnetic and magnetostrictive behaviors of the pseudobinary ferrimagnetic spinel oxide system (1−*x*)CoFe_2_O_4_–*x*CoAl_2_O_4_ [Co(Fe_1−*x*_Al_*x*_)_2_O_4_], with one end-member being the ferrimagnetic CoFe_2_O_4_ and the other end-member being CoAl_2_O_4_ that is paramagnetic above 9.8 K. The temperature spectra of magnetization and magnetic susceptibility were employed to detect the magnetic transition temperatures and to determine the phase diagram of this system. Composition dependent and temperature dependent magnetostrictive behaviors reveal an exotic phase boundary that separates two ferrimagnetic states: At room temperature and under small magnetic fields (∼500 Oe), Fe-rich compositions exhibit negative magnetostriction while the Al-rich compositions exhibit positive magnetostriction though the values are small (<10 ppm). Moreover, the compositions around this phase boundary at room temperature (*x* = 0.35, 0.4, 0.45, 0.5) exhibit near-zero magnetostriction and enhanced magnetic susceptibility, which may be promising in the applications for magnetic cores, current sensors, or magnetic shielding materials.

## 1. Introduction

As one main type of the ferrimagnetic functional materials, the ferrites with the spinel structure are widely used in the key components of sensors, transducers, and actuators [1]. Among the various spinel ferrites, cobalt ferrite (CoFe_2_O_4_) is well known for its high magnetostriction, large magneto-crystalline anisotropy, high corrosion resistance, good chemical stability, and low cost [2,3]. Therefore, CoFe_2_O_4_-based materials are good candidates for magneto-mechanical and sensor devices. 

Considering that CoFe_2_O_4_ possesses an inverse spinel structure in which the Co^2+^ ions are situated at the octahedral site (B-site) and the Fe^3+^ ions are situated in both the tetrahedral site (A-site) and the octahedral site [4], the moments of the octahedral sites and the tetrahedral sites align antiparallel and the net moment is the subtraction of the two. Thus, the cobalt ferrite and substituted cobalt ferrite are ferrimagnetic. The magnetic and mechanical properties of CoFe_2_O_4_ can be tuned by doping with other elements. Much effort has been dedicated to investigating the effects of doping other elements into CoFe_2_O_4_, including Ti^4+^, Mn^3+^, Cr^3+^, Ge^4+^, Ga^3+^, Si^4+^, etc. [5,6,7,8,9,10]. Nlebedim et al. reported that the substitution of Al^3+^ for Fe^3+^ could result in the decrease of magnetostriction and the change of magnetostrictive coefficient, thus making cobalt–ferrite materials qualified for the sensors and actuators [11,12]. The nanocrystalline Al-doped CoFe_2_O_4_ powders fabricated via the sol–gel auto-ignition method were also studied systematically, indicating that the nanocrystalline Al-doped CoFe_2_O_4_ is a good candidate for high frequency applications [13,14]. Anantharamaiah and Joy investigated the site preference of Al in CoFe_2_O_4_ and concluded that Al^3+^ substitutes Fe^3+^ at both crystallographic sites (A-site and B-site) [15]. 

Despite the intense interest on this system, the phase diagram of Al-doped CoFe_2_O_4_ is still lacking. In this work, based on the temperature spectrum of magnetic susceptibility within a wide temperature range (10–800 K), we obtained the phase diagram of (1−*x*)CoFe_2_O_4_–*x*CoAl_2_O_4_ [hereinafter Co(Fe_1−*x*_Al_*x*_)_2_O_4_, *x* varies from 0 to 1] for the first time and also discussed the relation between the magnetostrictive properties and the magnetic phase boundary. 

## 2. Materials and Methods 

The polycrystalline samples of Co(Fe_1−*x*_Al_*x*_)_2_O_4_ (*x* = 0.0, 0.05, 0.1, 0.15, 0.2, 0.25, 0.3, 0.35, 0.4, 0.45, 0.5, 0.6, 0.75, 0.9, 1.0) were prepared using a conventional solid-state reaction method, and the starting chemicals were Al_2_O_3_ (99.99%), Fe_2_O_3_ (99.9%), and CoO (95%) powders. The mixed powders according to the stoichiometry were ball milled for 3 h and then pre-sintered at 1000 °C for 24 h in air atmosphere. To achieve better homogeneity, the powders were re-milled for 6 h. Then, the powders were granulated using 10% polyvinyl alcohol (PVA) and then pressed into disks with thickness ∼0.1 cm and diameter ∼0.9 cm under a hydro-press of 20 MPa. Finally, the pressed disk samples were sintered at 1350 °C for 24 h in air atmosphere, followed by furnace cooling to room temperature. The elemental compositions were determined using energy-dispersive X-ray spectroscopy (EDX) in a scanning electron microscope (SEM, JEOL JSM-7000F, Tokyo, Japan). The phase purity of the calcined powder samples was examined by X-ray diffraction (XRD, Bruker D8 ADVANCE, Hamburg, Germany) using Cu Kα radiation with an angle (2θ) step of 0.02° between 15° and 85°. The lattice constants of the synthesized samples were calculated from XRD data. Magnetization (*M*) versus temperature (*T*) curves under a field of 100 Oe were measured using the superconducting quantum interference device (SQUID) over the temperature range of 300–800 K. The Curie temperature (T_C_) was determined from the *M* vs. *T* curve by linear extrapolation from the region of maximum slope down to the temperature axis. The temperature spectrum of magnetic susceptibility was measured using SQUID with the frequency of 133 Hz, the magnetic field of 0.1 Oe in the temperature range of 10–800 K. The magnetization (*M*) versus magnetic field (*H*) hysteresis loops were measured using the SQUID at 300 K. Magnetostriction was measured using the strain gage within a temperature range of 100–300 K, and the obtained magnetostriction was longitudinal magnetostriction (the applied magnetic field was parallel to the strain gauge). 

## 3. Results and Discussion

### 3.1. Composition Analysis and Structural Characterization

The Energy-dispersive X-ray spectroscopy (EDX) results are shown in Table 1. The detected elemental content agrees well with the nominal elemental content. Thus, hereinafter, the compositions mentioned in the manuscript are all nominal compositions.

The XRD patterns of some selected typical compositions are shown in Figure 1a. The XRD patterns demonstrate that the samples possess cubic spinel structure without any secondary phase, and with the increasing content of Al, all the characteristic reflections shift to higher angles, indicating that the lattice constant decreases monotonously with the increasing concentration of Al^3+^. 

Based on the XRD profile of the reflection (311) as shown in Figure 1b, the full width at half maximum (FWHM) of the intensity of the reflection (311) is employed to calculate the grain size using the Debye–Scherrer equation [13,16,17]:
D = 0.9λ/(Bcosθ),(1)
where D is grain size, λ is the X-ray wavelength, B is the FWHM of (311), and θ is the Bragg angle. The calculated grain size for each composition is shown in Figure 1c. With the increase of the Al^3+^ concentration, the grain size of the samples decreases, indicating that the doping of Al in CoFe_2_O_4_ hinders the grain growth. It is also observed that from *x* = 0.35 to *x* = 0.5, there is an abrupt decrease of the grain size. The abrupt change will be investigated later in another work. 

Figure 1d shows the calculated lattice constant for each composition. The radius of Al^3+^ is smaller than Fe^3+^ and Co^2+^, so with the increasing content of Al, the lattice constant decreases monotonously [15,18].

### 3.2. Temperature (T) Spectra of Magnetization (M) and Magnetic Susceptibility (χ)

The temperature (*T*) spectra of the magnetization (*M*) for the Co(Fe_1−*x*_Al_*x*_)_2_O_4_ samples is shown in Figure 2. The Curie temperature (T_C_) can be determined by the first derivative of the *M*–*T* curve. Since the magnetism of cobalt ferrite comes from the super-exchange interaction between the magnetic metal ions, the substitution of nonmagnetic Al^3+^ weakens the super-exchange interaction between the spinel A-site and B-site (J_ex_ of Fe–O–Fe > Co–O–Fe >> Fe–O–Al ≈ Co–O–Al) [19], which finally leads to the decrease of T_C_ with increasing concentration of Al.

Figure 3 shows the low-field a.c. susceptibility versus temperature curves for some selected compositions of Co(Fe_1−*x*_Al_*x*_)_2_O_4_. For pure CoFe_2_O_4_, the *χ*–*T* curve exhibits a sharp peak and a broad hump, indicating two transitions: The prominent peak refers to the Curie temperature, and the broad hump may refer to a ferrimagnetic-ferrimagnetic transition. The Curie temperatures detected from the magnetization versus temperature curves are consistent with those detected from the magnetic susceptibility versus temperature curves in Figure 3. Because of previous lack of high-temperature measurement data, the ferrimagnetic–ferrimagnetic transition around 550 K for CoFe_2_O_4_ was neither identified nor reported. With the increase of Al content, all the transition temperatures decrease. For the compositions of *x* = 0.25–0.5, the ferrimagnetic–ferrimagnetic transition is found to split into two transitions, which are labeled with black arrows in the figure.

With higher concentration of Al, *x* > 0.6, the peak value of magnetic susceptibility decreases sharply with increasing *x*, suggesting that the Curie transition becomes weaker; the Curie transition almost vanishes at *x* = 1.0 (CoAl_2_O_4_), which is reported to demonstrate a paramagnetic–antiferromagnetic transition at 9.8 K that is beyond the test temperature range in the current study [20]. 

### 3.3. Phase Diagram

Based on the transition temperatures determined from the temperature spectra of magnetization and magnetic susceptibility, the phase diagram of the Co(Fe_1−*x*_Al_*x*_)_2_O_4_ system is illustrated as Figure 4. Obviously, with the increasing content of Al, the phase transitions in CoFe_2_O_4_ gradually converge until the para–ferri transition almost vanishes at *x* = 1.0 (CoAl_2_O_4_) [20]. 

We notice that Supriya et al. studied the transition behavior in nanocrystalline Al-doped CoFe_2_O_4_ and discovered two magnetic transitions in Co(Fe_0.9_Al_0.1_)_2_O_4_ [21]. In the present study, the composition dependence of the two magnetic transition temperatures were investigated based on the response of the moment upon the magnetic field. As will be shown later in the Section 3.5, the magnetic transition from ferrimagnetic-A state to ferrimagnetic B state accompanies the change of the two magnetostriction coefficients λ_100_ and λ_111_. 

### 3.4. M–H Hysteresis Loops

Substitution of the nonmagnetic Al ion for Fe ion in cobalt ferrite has a strong impact on the magnetic properties as reflected in the magnetization (*M*) versus magnetic field (*H*) hysteresis loops (Figure 5). According to the previous experimental results, in CoFe_2_O_4_, half of Fe^3+^ ions take A-site, Co^2+^ and the other half of Fe^3+^ ions take B-site [22]. When Al^3+^ is doped into CoFe_2_O_4_, it has no preference on either A-site or B-site [23]. Therefore, no matter the Al^3+^ ions take A-site or B-site, the replacement of Fe^3+^ with Al^3+^ decreases the super-exchange interaction between the sub-lattices and then decreases the magnetization [14]. Meanwhile, the coercive field (H_C_) also decreases with increasing Al content. The variation of H_C_ is associated with the changes in the magnetocrystalline anisotropy [19]. The large coercive field of CoFe_2_O_4_ originates from the magnetocrystalline anisotropy of the cobalt ions at the B-sites caused by its spin–orbit coupling [24]. As the concentration of Al ions increases, the magnetocrystalline anisotropy of Co(Fe_1−*x*_Al_*x*_)_2_O_4_ decreases, thus the H_C_ decreases [11,12,19,25].

### 3.5. Magnetostriction as a Function of Composition and Temperature

The magnetostriction (*λ*) versus magnetic field (*H*) curves at 300 K for the Co(Fe_1−*x*_Al_*x*_)_2_O_4_ system are shown in Figure 6a. In general, the magnetostriction decreases as the Al content increases. For cobalt ferrite materials, the smaller the grain size, the larger the magnetostriction [4]. For Al-doped CoFe_2_O_4_ polycrystalline samples in the present study, with the decrease of grain size, the saturated magnetostriction decreases. This indicates that the doping of Al rather than sintering process (grain size) plays the dominant role in the change of magnetostriction.

For the compositions of *x* = 0–0.3, the magnetostriction first increases to a maximum negative value and then decreases to saturation; the saturated negative magnetostriction of these compositions decreases with the increasing content of Al. For the compositions of *x* = 0.4–0.5, the magnetostriction first increases to a positive value and then decreases to negative. Such sign change of magnetostriction at the low magnetic field (≤500 Oe) indicates the change of both the magnetostriction coefficient λ_100_ and λ_111_ [11,12,25]. The CoFe_2_O_4_ has two magnetostriction coefficients λ_100_ and λ_111_: λ_100_ < 0 and λ_111_ > 0 at 300 K [2]. Since the easy magnetization axis of CoFe_2_O_4_ is [100], correspondingly, it has a large negative λ_100_ and a small positive λ_111_ [12]. As the content of Al^3+^ increases, the sign of the magnetostriction under the low field changes from negative to positive, which suggests that the contribution of λ_100_ becomes less and the contribution of λ_111_ becomes dominant. As more Al^3+^ replaces Fe^3+^, the magnetocrystalline anisotropy decreases and results in smaller saturated magnetostriction [11,12]. Among all the compositions, *x* = 0.35 shows the minimum saturated magnetostriction (~−5 ppm). The magnetostriction at low field can be referred to Figure 6b.

Figure 6c shows the temperature dependent magnetostriction curves of *x* = 0.35, 0.4, 0.45, and 0.5. For *x* = 0.35, from 150 to 300 K, it exhibits negative magnetostriction and there is no sign change for the magnetostriction; for *x* = 0.4, 0.45, and 0.5, they all show sign changes of magnetostriction under small fields at certain temperatures. It should be noted that the sign changing temperature agrees well with the detected transition temperature for *x* = 0.4, 0.45, and 0.5.

The temperature and composition dependence of magnetostriction for Co(Fe_1−*x*_Al_*x*_)_2_O_4_ makes it possible to design desired compositions for applications of interest. Another work will follow to verify whether the solution system of a ferrimagnetic system and a paramagnetic system can form the phase boundary where the composition shows enhanced magnetic susceptibility and weakened magnetostriction, as shown in Figure 7. It should be noted that such phenomena have been reported to exist in the ferromagnetic morphotropic phase boundary (MPB) involved system [26]. 

## 4. Conclusions

In conclusion, we fabricated a pseudo-binary spinel oxide system Co(Fe_1−*x*_Al_*x*_)_2_O_4_ (*x* = 0–1) that covers the whole composition range and determined its phase diagram for the first time. As the concentration of nonmagnetic Al^3+^ increases, the lattice constant, Curie temperature T_C_, saturated magnetization, and coercive field of polycrystalline Co(Fe_1−*x*_Al_*x*_)_2_O_4_ samples all decrease monotonously. Moreover, it was found that the ferrimagnetic–ferrimagnetic phase boundary in the phase diagram (Figure 4) distinguishes two ferrimagnetic states with different λ_100_ and λ_111_. Around the ferrimagnetic–ferrimagnetic phase boundary, the compositions of *x* = 0.35, 0.4, 0.45, and 0.5 exhibit near-zero magnetostriction and enhanced magnetic susceptibility, which can be utilized to design new magneto-responsive materials that require high susceptibility and low strain.

## Figures and Tables

**Figure 1 materials-12-01685-f001:**
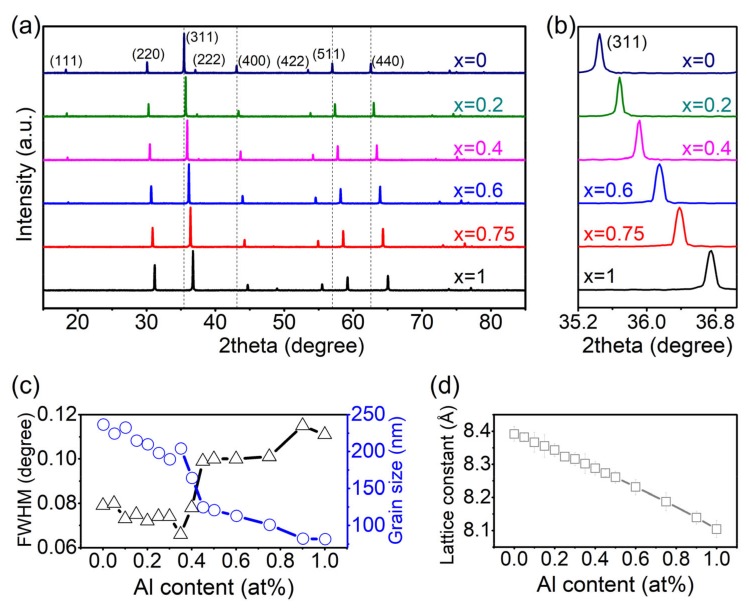
(Color online) (**a**) XRD patterns for some selected compositions Co(Fe_1−*x*_Al_*x*_)_2_O_4_ (*x* = 0, 0.2, 0.4, 0.6, 0.75, 1); the dashed lines are guidance for the shift of reflections; (**b**) the (311) reflections of the XRD profiles in (**a**); (**c**) the composition dependence of full width at half maximum (FWHM) and the calculated grain size; (**d**) the composition dependence of lattice constant.

**Figure 2 materials-12-01685-f002:**
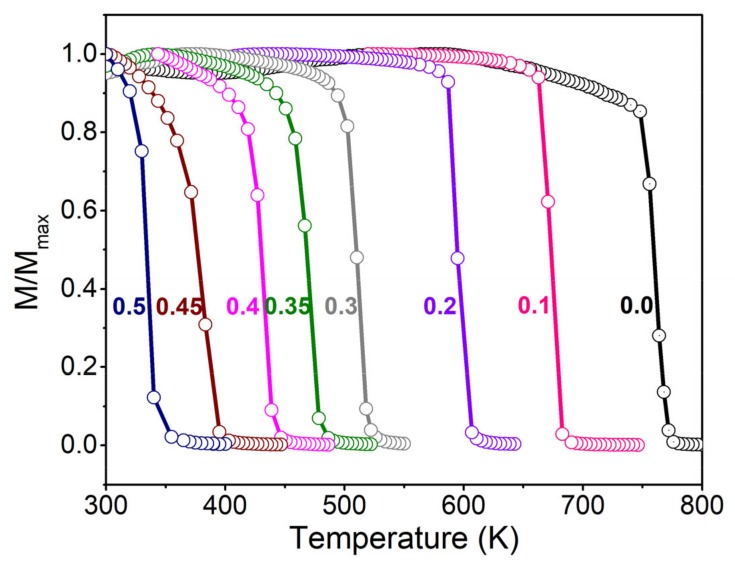
(Color online). The normalized magnetization versus temperature curves for Co(Fe_1−*x*_Al_*x*_)_2_O_4_ (*x* = 0.0, 0.1, 0.2, 0.3, 0.35, 0.4, 0.45, 0.5).

**Figure 3 materials-12-01685-f003:**
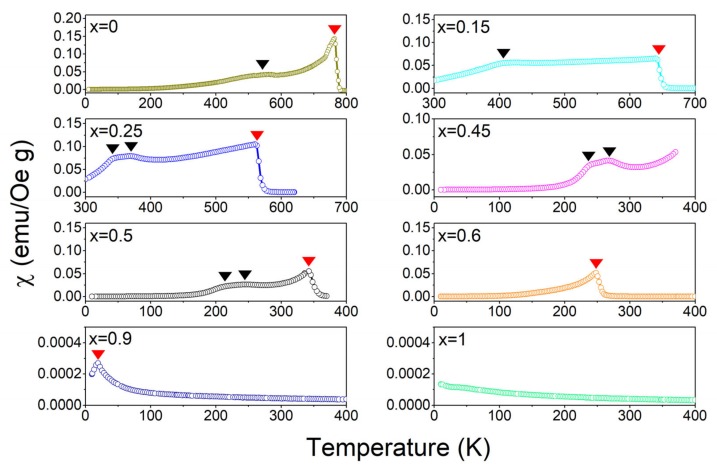
(Color online). The magnetic susceptibility versus temperature curves for Co(Fe_1−*x*_Al_*x*_)_2_O_4_ (*x* = 0.0, 0.15, 0.25, 0.45, 0.5, 0.6, 0.9, 1.0). The red triangle denotes the Curie transition, and the black ones denote the ferrimagnetic-ferrimagnetic transition.

**Figure 4 materials-12-01685-f004:**
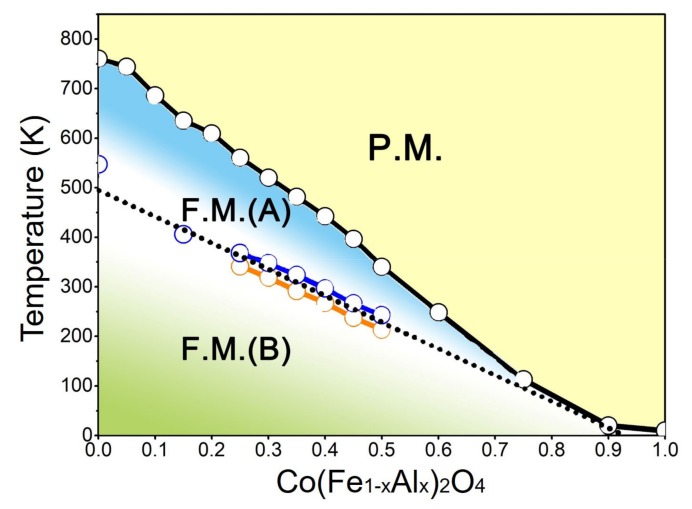
(Color online). The temperature–composition phase diagram of Co(Fe_1−*x*_Al_*x*_)_2_O_4_. P.M. denotes paramagnetic phase; F.M. (A) and F.M. (B) denote the ferrimagnetic A phase and ferrimagnetic B phase, respectively. Red circles and blue circles refer to the anomalies appearing on the magnetic susceptibility versus temperature curves in Figure 3.

**Figure 5 materials-12-01685-f005:**
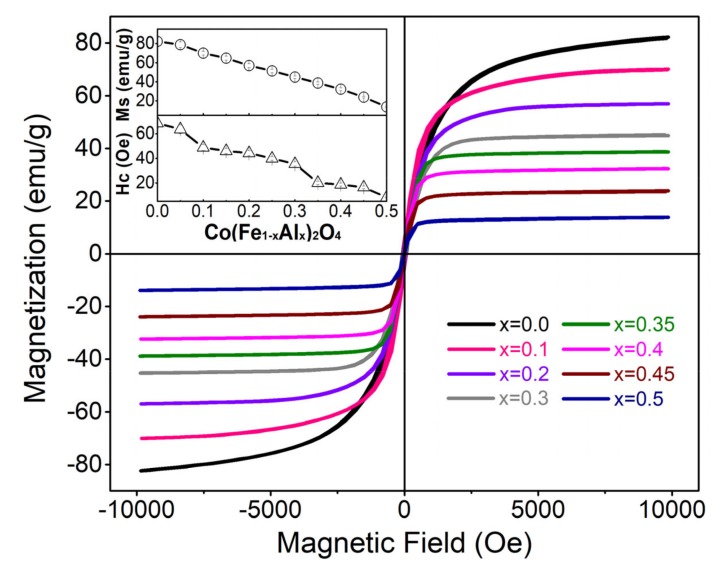
(Color online). The magnetization curves of Co(Fe_1−*x*_Al_*x*_)_2_O_4_ (*x* = 0.0, 0.1, 0.2, 0.3, 0.35, 0.4, 0.45, 0.5) at room temperature with the insets showing the saturated magnetization and coercive field for each composition.

**Figure 6 materials-12-01685-f006:**
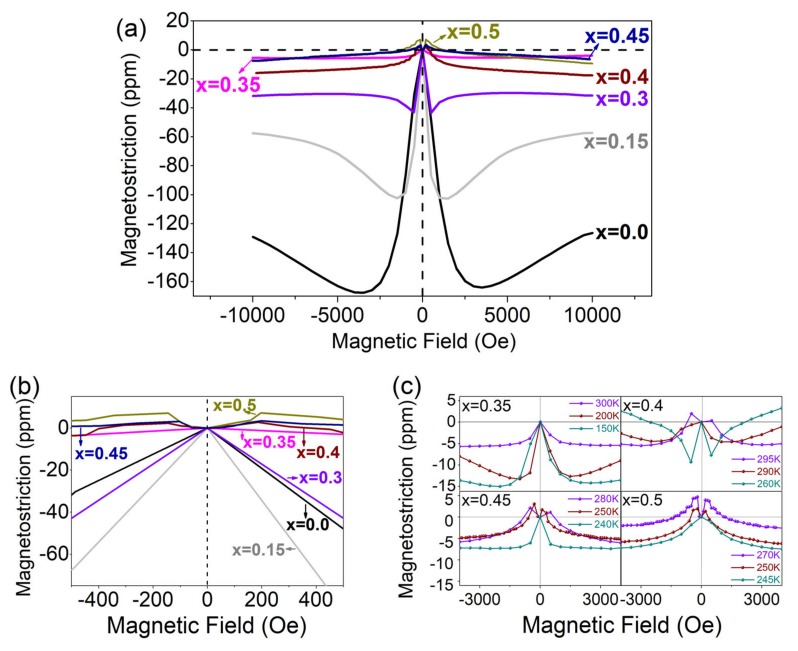
(Color online). (**a**) Magnetostriction curves of Co(Fe_1−*x*_Al_*x*_)_2_O_4_ (*x* = 0.0, 0.15, 0.3, 0.35, 0.4, 0.45, 0.5) at 300 K; (**b**) the magnetostriction curves under 500 Oe; (**c**) temperature dependent magnetostriction curves of Co(Fe_1−*x*_Al_*x*_)_2_O_4_ (*x* = 0.35, 0.4, 0.45, 0.5).

**Figure 7 materials-12-01685-f007:**
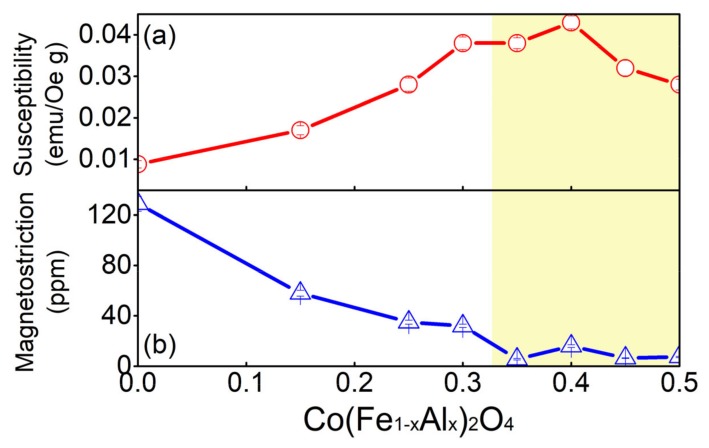
(Color online). Composition dependence of magnetic susceptibility (**a**) and saturated magnetostriction (**b**) at room temperature. The yellow-shadowed region denotes the composition range that exhibits high magnetic susceptibility and low magnetostriction.

**Table 1 materials-12-01685-t001:** The nominal composition and the elemental content detected from the EDX.

Nominal Composition	Composition by EDX Analysis			
	Co	Fe	Al	O
CoFe_2_O_4_	14.2%	27.7%	0	58.1%
Co(Fe_0.95_Al_0.05_)_2_O_4_	14.3%	26.9%	1.3%	57.6%
Co(Fe_0.85_Al_0.15_)_2_O_4_	14.5%	24.5%	4.1%	56.9%
Co(Fe_0.8_Al_0.2_)_2_O_4_	13.4%	21.3%	5.7%	59.6%
Co(Fe_0.75_Al_0.25_)_2_O_4_	13.4%	19.7%	7.1%	59.8%
Co(Fe_0.7_Al_0.3_)_2_O_4_	13.7%	18.0%	7.7%	60.6%
Co(Fe_0.65_Al_0.35_)_2_O_4_	13.4%	17.2%	9.8%	59.6%
Co(Fe_0.6_Al_0.4_)_2_O_4_	12.5%	16.0%	10.7%	60.8%
Co(Fe_0.55_Al_0.45_)_2_O_4_	13.5%	15.8%	12.6%	58.0%
Co(Fe_0.5_Al_0.5_)_2_O_4_	14.5%	12.5%	13.2%	59.8%
Co(Fe_0.4_Al_0.6_)_2_O_4_	14.2%	10.3%	17.1%	58.5%
Co(Fe_0.25_Al_0.75_)_2_O_4_	14.7%	6.7%	19.3%	59.4%
Co(Fe_0.1_Al_0.9_)_2_O_4_	15.0%	2.2%	23.4%	59.4%
CoAl_2_O_4_	13.5%	0	25.7%	60.8%

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
