# Peer review of "The Phase Diagram and Exotic Magnetostrictive Behaviors in Spinel Oxide Co(Fe1−xAlx)2O4 System"

_materials, 2019, doi:10.3390/ma12101685_

Reviewer 1 Report

Dear Editor,

I reviewed the resubmitted manuscript “The Phase Diagram and Exotic Magnetostrictive Behaviors in Spinel Oxide Co(Fe1-xAlx)2O4 System” by Chao Zhou, Azhen Zhang, Tieyan Chang, Yusheng Chen, Yin Zhang, Fanghua Tian, Wenliang Zuo, Yang Ren, Xiaoping Song, and Sen Yang. The motivation and the literature review (my main concern about the previous version) have been improved in the reworked manuscript. The main result now is formulated as the magnetic phase diagram in the wide temperature and composition ranges. It seems that such a phase diagram was not reported before and can be considered as a new result on this particular material. At that the manuscript still needs improvements in my opinion. Authors should clarify some points prior to my recommendation. Below please, find my questions.

 1)    Authors mention that two phases were reported in Ref. [21]. They also state that these phases were obtained using conductivity measurements and therefore are not directly related to the phases obtained in the present manuscript using magnetic measurements.

 I would like to note that in Ref. [21] magnetic susceptibility were also reported. So, two magnetic phases were observed previously. This should be mentioned in the manuscript. However, I agree that the magnetic measurements in Ref. [21] were performed for only a single Al concentration.

 2)    It is not clear how author define the type of magnetic structure in the system. They mention paramagnetic-ferrimagnetic (a) and ferrimagnetic (a)-ferrimagnetic (b) transitions. How do they know that the system is ferrimagnetic but not ferromagnetic (for example)?

 By the way FM(a) and FM(b) notations are slightly confusing in Fig. 4. My first guess is that they mean ferromagnetic phase a and b, but not the ferromagnetic phases.

 3)    Authors claim that change of magnetostriction sign occurs around the phase transition. This is shown for only few Al concentrations (0.4, 0.45 and 0.5). Did they measure and observe similar sign change around the phase transition for other concentrations? Without such measurements it is better to avoid making such a general statement in the paper conclusion.

       4)      Numbers indicating temperatures in Fig. 6c, Al content in Fig. 6b and magnetic field in         the inset in Fi. 4 are too small.

          5)      The line indicating zero magnetostriction in Fig. 6c would be useful.

Author Response

Point 1: Authors mention that two phases were reported in Ref. [21]. They also state that these phases were obtained using conductivity measurements and therefore are not directly related to the phases obtained in the present manuscript using magnetic measurements.

I would like to note that in Ref. [21] magnetic susceptibility were also reported. So, two magnetic phases were observed previously. This should be mentioned in the manuscript. However, I agree that the magnetic measurements in Ref. [21] were performed for only a single Al concentration.

 Response 1: Thanks for the reviewer’s reminding. In the revised version, we made corresponding modifications in the revised version (Page6, Lines146-151).

 Point 2: It is not clear how author define the type of magnetic structure in the system. They mention paramagnetic-ferrimagnetic (a) and ferrimagnetic (a)-ferrimagnetic (b) transitions. How do they know that the system is ferrimagnetic but not ferromagnetic (for example)?

By the way FM(a) and FM(b) notations are slightly confusing in Fig. 4. My first guess is that they mean ferromagnetic phase a and b, but not the ferromagnetic phases.

 Response 2: Thanks for the reviewer’s comment. We should but did not introduce the magnetic structure of cobalt ferrite. There are two sub-lattices in the cobalt ferrite, one is tetrahedral and the other one is octahedral. The moments of the two sub-lattices (the octahedral sites and the tetrahedral sites) align antiparallel and the net moment is the subtraction of the two. Therefore, the cobalt ferrite and the substituted cobalt ferrite (as in the present study) are ferrimagnetic. In the revised version, we made corresponding modifications (Page1, Lines38-40).

For the F.M.(a) and F.M.(b), we indeed mean ferrimagnetic a and b, we make it clear in the revised version (Page6, Lines143-144).  

 Point 3: Authors claim that change of magnetostriction sign occurs around the phase transition. This is shown for only few Al concentrations (0.4, 0.45 and 0.5). Did they measure and observe similar sign change around the phase transition for other concentrations? Without such measurements it is better to avoid making such a general statement in the paper conclusion.

 Response 3: Thanks for the reviewer’s comment. Limited by the test instrument, the temperature limit for the measurement of magnetostriction is 50 K ~ 320 K. Therefore, based on the consistence between the sign-changing temperature (Fig.6) and the detected transition temperature (Fig.4), it can be predicted that the sign of magnetostriction changes around the phase transition for all compositions. However, as suggested by the reviewer, considering the fact that we could not provide the data above 320 K, for the sake of accuracy, we made corresponding modifications (Page7/8, Lines197-201).

 Point 4: Numbers indicating temperatures in Fig. 6c, Al content in Fig. 6b and magnetic field in the inset in Fi. 4 are too small.

 Response 4: Thanks for the reviewer’s suggestion. We have made corrections accordingly.

 Point 5: The line indicating zero magnetostriction in Fig. 6c would be useful.

 Response 5: Thanks for the reviewer’s comment. We have changed the dashed lines to solid lines to indicate the zero magnetostriction in Fig. 6c.

Reviewer 2 Report

The authors have improved their manuscript and replied to the referees' comments accordingly. Still the English can be improved and its noted that there are some typos in the manuscript which should be polished before the publication.

Author Response

Point 1: The authors have improved their manuscript and replied to the referees' comments accordingly. Still the English can be improved and it is noted that there are some typos in the manuscript which should be polished before the publication.

 Response 1: Thanks for the reviewer’s comment. We have improved some sentences and corrected the typos in the revised version.

Reviewer 3 Report

The authors have significantly improved the manuscript. I recommend the manuscript for publication.

My only remaining concern is the number of digit in the EDX data. Instead of writing "It should be pointed out that the resolution of EDX cannot reach 0.0x%, and the detected compositions in Table 1 can only provide." , I recommend to delete this sentence and display only one instead of two digits. It does not make sense to show more digits than the error/uncertainty allows.

Author Response

Point 1: The authors have significantly improved the manuscript. I recommend the manuscript for publication. My only remaining concern is the number of digit in the EDX data. Instead of writing "It should be pointed out that the resolution of EDX cannot reach 0.0x%, and the detected compositions in Table 1 can only provide." , I recommend to delete this sentence and display only one instead of two digits. It does not make sense to show more digits than the error/uncertainty allows.

 Response 1: Thanks for the reviewer’s suggestion. We have made corrections accordingly. 

Round  2

Reviewer 1 Report

Dear Editor,

Authors introduced necessary corrections to their manuscript. In my opinion it can be published now.